# Effects of Different Bacteriostats on the Dynamic Germination of *Clostridium perfringens* Spores

**DOI:** 10.3390/foods12091834

**Published:** 2023-04-28

**Authors:** Dong Liang, Shengnan Liu, Miaoyun Li, Yaodi Zhu, Lijun Zhao, Lingxia Sun, Yangyang Ma, Gaiming Zhao

**Affiliations:** 1College of Food Science and Technology, Henan Agricultural University, Zhengzhou 450002, China; liangdong0221@126.com (D.L.); 18839004375@163.com (S.L.);; 2International Joint Laboratory of Meat Processing and Safety in Henan Province, Henan Agricultural University, Zhengzhou 450002, China

**Keywords:** *Clostridium perfringens*, bacteriostat, spores, germination

## Abstract

Bacteriostats, as chemical substances that inhibit bacterial growth, are widely used in the sterilization process; however, their effects on spindle spores are unclear. In this study, the effects of bacteriostats, including nine commonly used food additives and four detergents, on the growth of *Clostridium perfringens* spores were investigated. The results showed that 0.07‰ ethylenediaminetetraacetate had a good inhibitory effect on *C. perfringens* spore growth, and the spore turbidity decreased by 4.8% after incubation for 60 min. Furthermore, 0.3‰ tea polyphenols, 0.8‰ D-isoascorbic acid, and 0.75‰ potassium sorbate promoted leakage of contents during spore germination. Among the four detergents, 5‰ glutaraldehyde solution presented the best inhibitory effect on the growth of *C. perfringens* spores, and the spore turbidity decreased by 5.6% after incubation for 60 min. Further analysis of the inactivation mechanism of spores by the bacteriostats was performed by comparing the leakage of UV-absorbing substances during germination. The results revealed that bacteriostats could not directly kill the spores, but could inactivate them by inhibiting germination or damaging the spore structure during germination, thus preventing the formation of bacterial vegetative bodies. These findings provide important information and reference for the mechanism underlying the effects of different bacteriostatic agents on spore growth.

## 1. Introduction

*Clostridium perfringens* (*C. perfringens*) is a common food-borne bacterium that can form spores in harsh growth environments. These bacterial spores are widely distributed in air, soil, sewage, and food, and are difficult to kill owing to their unique structure [1,2,3]. Under suitable conditions for bacterial growth, the spores germinate and become vegetative forms, causing food contamination and bag swelling, resulting in serious economic losses and threatening the health and safety of consumers [4,5,6,7]. In the current food production process, bacteriostatic agents are commonly added to raw food materials and auxiliary materials, and cleaning agents are sprayed in the production environment to eliminate and kill harmful bacteria, thereby preventing food spoilage, prolonging the shelf life of food, and ensuring food safety [8,9,10,11].

In recent years, the effects of a variety of chemical and physical components on the germination and expansion of spores have been well investigated. Some studies have shown that nisin could inhibit the germination and reproduction of heat-activated spores, and that ethylene oxide could inactivate proteins and nucleic acids by reacting with different chemical groups, essentially destroying spores [12]. However, it is difficult to identify basic pressure breaking as the primary cause of spore death [3], and the changes in the external environment pH, osmotic pressure, temperature, chemical treatment, etc., are insufficient to completely kill the spores [13]. Nevertheless, once the spores germinate and become vegetative forms, the bacteriostatic agent can enter the thalli and cause acidification or extraction, possibly leading to metabolic disorders and cell death [11,14,15]. Many antibacterial agents have been shown to be effective against most of the Gram-positive and Gram-negative bacteria by destroying the cell wall or cell membrane, acting on intracellular nucleic acids or proteins, affecting their expression and synthesis, and inhibiting the activity of related enzymes in the cells [10,11,15,16]. For example, ԑ-polylysine hydrochloride, nitrite and other substances can destroy the integrity of the bacterial cell membrane and are thus selected as food preservatives [17]. Potassium sorbate inhibits the activity of microbial dehydrogenase and destroys multiple major enzyme systems to achieve preservative efficacy [11,16,18]. D- isoascorbic acid could reduce the oxygen concentration in the system and inhibit bacterial propagation [10]. "Tea polyphenols" is the general name for phenols in tea, including flavanols (catechins), flavonols and their predecessors, flavonoids (anthoxanthin), hydroxy-4-flavanols, anthocyanins, phenolic acids, phenolic acids, and other polyphenols [19]. Since tea polyphenols can change the normal morphology of bacteria and damage the cell wall structure, they have gradually become a natural antibacterial agent and are widely used [9]. Whey protein baking powder is a natural fresh-keeping food raw material. Unlike traditional bacteriostatic agents, it can be used in various foods to maximize the maintenance of food freshness and shelf life and block the spread of food-borne diseases. However, there is relatively little research on whey protein baking powder at present, and its antibacterial mechanism is still unclear. Glutaraldehyde is a high-efficiency disinfectant commonly used in industry [12]. Its bactericidal mechanism is mainly through the protein alkylation reaction of microorganisms, blocking the outer layer of cells of bacteria, and inactivating cell enzymes to kill bacteria. Trichloroisocyanuric acid and hydrogen peroxide Ag+ are industrial disinfectants with certain corrosivity [20]. Under certain conditions, they can be decomposed into small molecular substances and destroy the cell structure. Therefore, they are used as a new generation of high-efficiency disinfectants [12]. However, it is still unclear whether these agents can produce the same effects on spore growth. Studies have indicated that the resistance of spores varies with the chemicals [21]. Some oxidants have been found to damage the inner membrane of the spores, rupturing the inner membrane during germination [22], while other compounds, such as nitrous acid and formaldehyde, can kill spores through DNA damage [7,23]. Moreover, the mechanism by which some compounds kill spores is still unknown. While the majority of previous studies have focused on Bacillus spores, research on *Clostridium* spores, and particularly the wild-type spores from industry, is still limited. Moreover, the effect of a single antibacterial treatment on spore germination has only rarely been recorded [21,24,25].

The objective of this study was to test the inhibition and killing mechanisms of different bacteriostats on the representative strain of wild-type spores, *C. perfringens*. Spores were studied by comparing the effects of different microbial agents on spore germination and growth.

## 2. Materials and Methods

### 2.1. Materials and Reagents

Sodium dichloroisocyanurate, trichloroisocyanuric acid, 4-chloro-3,5-dimethylphenol, McLean Biochemical Technology Co., Ltd., Shanghai, China; terbium chloride(TbCl_3_), Nycodenz, tea polyphenols (catechin content accounted for 41.31%, epicatechin content accounted for 19.72%, quercetin content accounted for 3.71%, and gallic acid content accounted for 4.75%), Macklin Biochemical Technology Co., Ltd., Shanghai, China; nisin, nitrite, potassium sorbate, D-isoascorbic acid, sodium diacetate, ԑ-polylysine hydrochloride, Sigma-Aldrich Trading Co., Ltd., Shanghai, China; tryptose Sulfite cycloserine Agar Base (TSC), thioglycollate medium (FTG), Hi-tech Industrial Park Haibo Biotechnology Co., Ltd., Shandong, China; whey protein fermentation powder, yinong biology science and technology Co., Ltd., Shanghai, China; Other reagents such as glutaraldehyde, silver nitrate standard solution and hydrogen peroxide solution are analytically pure in China.

### 2.2. Instrumentation and Equipment

SpectraMax M2e multifunctional microplate reader, Finland, California USA; MIR-254 Low-temperature Incubator, SANYO Company, Osaka, Japan; SW-CJ-2F clean bench, Suzhou Aetna Air Technology Co., Ltd., Jiangsu, China; Vortex-2 Genee Vortex Oscillator, ScientificIndustries, CA, USA; ECLIPSE 80i Biological Microscope, NIKON, Osaka, Japan; FA 2004A Electronic Analytical Balance, Jingtian Electronic Instrument Co., Ltd., Shanghai, China; HVE-50 steam autoclave, HIRAYAMA Co., Ltd., Osaka, Japan; Fluorescence Spectrometry, Thermo Fisher Scientific Shile Technology (China) Co., Ltd., Shanghai, China; ALLEGRA-64A high-speed refrigerated centrifuge Beckman Coulter, USA; HH-501 digital display super constant temperature water bath, Jintan Jerry Electric Appliance Co., Ltd., Changzhou, China.

### 2.3. Preparation of C. perfringens Suspension and Spores

In this study, a wild-type *C. perfringens* was used, which was isolated directly from a vacuum-packaged cooked meat by the Microbiology Laboratory of the College of Food Science and Technology, Henan Agricultural University (China) and identified by Sangon Biotech Co., Ltd., (Shanghai, China).

To prepare *C. perfringens* suspension, typical black colonies of *C. perfringens* on TSC plates were selected and inoculated into fresh FTG medium for anaerobic cultivation at 37 °C for 18 h until growth stability stage, according to Ren’s method [26]. For anaerobic cultivation at 37 °C for 48 h, the usual black colonies of *C. perfringens* chosen from TSC plates were inoculated into 10 mL of cornstarch medium. The *C. perfringens* spores were prepared as described previously [27], and the spore suspension was centrifuged at 5000× *g* and 4 °C for 10 min, washed with sterile water 5–7 times, suspended in sterile peptone water, and stored at 20 °C until further use.

### 2.4. Treatment of Spore Suspension with Different Bacteriostats

A known standard of spore suspension and a bacteriostat were added to a sterile polytetrafluoroethylene centrifuge tube. The concentrations of food additive bacteriostats, including tea polyphenols, nisin, nitrite, potassium sorbate, D-isoascorbic acid, sodium diacetate, ԑ-polylysine hydrochloride, whey protein fermentation powder, and ethylenediaminetetraacetate, were adjusted to 0.3‰, 0.5‰, 0.15‰, 0.75‰, 0.8‰, 3‰, 5%, 1.5‰, and 0.07‰ in water, according to the allowable maximum addition amount of the national standard for food safety, respectively, while the concentrations of detergent bacteriostats, including 4-chloro-3,5-dimethylphenol, hydrogen peroxide Ag+, sodium dichloroisocyanurate and trichloroisocyanuric acid mixed solution, and glutaraldehyde, were adjusted to 0.45‰, 0.5‰, 1.2‰, and 5‰, respectively, according to industrial production standards [28].

### 2.5. Determination of Antibacterial Activity

A total of 4–5 rings of *C. perfringens* were picked out by inoculation ring and inoculated into FTG medium at 37 °C for 12 h. Subsequently, *C. perfringens* suspension containing cells at a concentration of 10^8^ CFU/mL (optical density (OD_600_) 0.8–1.0) was prepared using sterilized normal saline, and 100 µL of the bacterial suspension was evenly inoculated onto a sterile plate with Oxford cup in the center of the plate. The Oxford cup contained 100 µL of the bacteriostat. The blank control comprised distilled water. All of the plates were incubated in a constant-temperature incubator at 37 °C for 12 h. After incubation, the Oxford cup was taken out for observation, and the size of the bacteriostatic ring was measured and recorded [29].

### 2.6. Determination of Spore Survival Rate

The spore samples were incubated with a bacteriostat at 37 °C for 12 h in a water bath, gradient-diluted, and then incubated in TSC medium at 37 °C for 24 h for enumeration, and the viable number of *C. perfringens* spores was determined by the plate counting method.

### 2.7. Determination of Spore Turbidity

A total of 200 µL of *C. perfringens* spore suspension was collected every 5 min for 1 h, and the spore turbidity (OD_600_%) was determined at 600 nm (Equation (1)), as described previously [30].
(1)$${\mathrm{OD}}_{600}\%\;=\frac{D_{d}}{D_{i}}\times 100,$$
where OD_600_% is the rate of change of OD_600_, D_d_ is the decreasing value of OD_600_, and D_i_ is the initial OD_600_ value.

### 2.8. Determination of 2,6-Pyridinedicarboxylic Acid Release from Spores

To determine 2,6-pyridinedicarboxylic acid (DPA) release from the spores, the spore suspension was centrifuged at 5180× *g* and 4 °C for 4 min, and the supernatant was treated with 50 µmol TBCl_3_ (pH = 5.6) at a ratio of 1:3. Subsequently, the fluorescence intensity of the sample was measured using a fluorescence spectrophotometer at an excitation wavelength of 270 nm and emission wavelength of 545 nm [31]. The untreated sample served as the negative control, and the sample that was heated at 121 °C for 20 min served as the positive control. The DPA release was calculated using Equation (2) as follows:(2)$$\mathrm{DPA}\;\left(\%\right)=\frac{F_{1}-F_{0}}{F_{2}-F_{0}}\times 100$$
where F_0_, F_1_, and F_2_ are the fluorescence intensities of untreated samples, bacteriostat-treated samples, and positive control samples, respectively.

### 2.9. Determination of Leakage of UV-Absorbing Substances from Spores

After treatment with various bacteriostats, the spore suspension was centrifuged at 4500× *g* for 15 min at 4 °C, and the supernatant was incubated with a germination inducer at 37 °C for 1 h. Subsequently, the absorbance of the sample was measured at 260 nm (for nucleic acid) and 280 nm (for protein) [32]. The untreated bacterial suspension was used as the control group, and sterile water was utilized as the blank control.

### 2.10. Determination of Refractive Power

To ascertain the refractive power of the spores treated with different bacteriostats, a few drops of spore suspension were placed onto a slide, covered with a coverslip, and observed for 12 h under a phase contrast microscope.

### 2.11. Statistical Analysis

There were at least three independent replicates, with two samples per replicate, for each treatment. Average microbial reductions, expressed in logarithmic values, were used in the statistical analyses. Comparisons between several groups were performed by one-way analysis of variance (ANOVA) using SPSS (version 26.0, Norman H. Nie, CA, USA), and the statistically significant differences were determined by Tukey’s post hoc analysis to analyze mean differences. Differences at *p* < 0.05 were considered significant. Graphs were constructed using Origin 2021 software.

## 3. Results

### 3.1. Bacteriostatic Activity of Different Bacteriostats

The commonly used food additives (including tea polyphenols, nisin, nitrite, potassium sorbate, D-isoascorbic acid, sodium diacetate, ԑ-polylysine hydrochloride, whey protein fermentation powder, and ethylenediaminetetraacetate) and detergents (such as 4-chloro-3,5-dimethylphenol, hydrogen peroxide Ag+, mixed solution of sodium dichloroisocyanurate and trichloroisocyanuric acid, and glutaraldehyde) were selected in this study to examine their antibacterial activity against *C. perfringens*. The diameter of the inhibition zone was used to assess the antibacterial activity of the bacteriostat. An inhibition zone with a diameter of >9 mm indicated antibacterial activity of the bacteriostat, and the larger the diameter of the inhibition zone, the stronger was the antibacterial activity of the bacteriostat [23]. Subsequently, the inhibitory effects of various bacteriostats on *C. perfringens* were compared, and the outcomes are presented in Table 1. The inhibitory effects of several food additives on *C. perfringens* significantly varied. The diameter of the inhibition zone for ethylenediaminetetraacetate, which had the best inhibitory effect on *C. perfringens* growth, was 35.33 mm, followed by those for nitrite and D-erythorbic acid, which were 32.53 and 24.67 mm, respectively. In contrast, whey protein fermentation powder did not impede the growth of *C. perfringens* vegetative cells. All four detergents could effectively inhibit *C. perfringens* growth. The diameter of the inhibition zone for the mixed solution of sodium dichloroisocyanurate and trichloroisocyanuric acid reached 30.27 mm. The antibacterial effect of the four detergents presented the following trend: glutaraldehyde > sodium dichloroisocyanurate and trichloroisocyanuric acid > hydrogen peroxide Ag+ > 4-chloro-3,5-dimethylphenol. 

### 3.2. Effects of Different Bacteriostats on Spore Inactivation

Spore viability was characterized by the number of colonies formed on the plate after a 24-h culture of the spores (Figure 1). When compared with the untreated spores, treatment with tea polyphenols exhibited the highest spore inactivation rate, reducing the number of *C. perfringens* spores by 5.015 log. Furthermore, treatment with potassium sorbate and D-isoascorbic acid caused a 3.222 log and 3.327 log decrease in spore vitality, respectively. Treatment with the four detergents dramatically decreased the survival rate of *C. perfringens* spores, and the following trend was observed: glutaraldehyde > hydrogen peroxide Ag+ > sodium dichloroisocyanurate and trichloroisocyanuric acid mixture > 4-chloro-3,5-dimethylphenol. Several bacteriostats exhibited good spore inactivation effects, as indicated by the plate counting assay. Although antibacterial agents are known to destroy spores before or after germination, the underlying mechanism is still unclear. Hence, in the present study, to better investigate the mechanisms of spore inactivation by bacteriostats, the precise stage of spore inactivation was identified by monitoring the dynamics of spore germination.

### 3.3. Effects of Different Bacteriostats on Spore Germination

#### 3.3.1. Effects of Different Bacteriostats on the Turbidity of Spores

The *C. perfringens* spores treated with various bacteriostats were induced in FTG medium to study the effects of these bacteriostats on the dynamics of spore germination. During spore germination, the cores expand by absorbing water, thus altering the refractive index. The OD_600_ value of the spore suspension presented a downward trend, and the brightness of the spores gradually decreased under phase contrast microscope [18,33] (Figure 2). The turbidity of spores treated with various antimicrobial agents did not significantly change after incubating for 20 min. After incubation for 60 min, significant differences were found between the turbidity of spores treated with ethylenediaminetetraacetate, nisin, tea polyphenols, D-isoascorbic acid, and Ꜫ-polylysine hydrochloride and the untreated spores. While the OD_600_ value of the untreated spore suspension decreased by 13.3% after 60 min, that treated with ethylenediaminetetraacetate did not present a significant decrease, indicating that ethylenediaminetetraacetate could inhibit germination and growth of *C. perfringens* spores. The turbidity of the spore suspension treated with glutaraldehyde decreased by 5.53%, when compared with those of untreated spores and spores treated with other detergents, indicating that glutaraldehyde inhibited germination of spores and prevented the formation of vegetative bodies.

#### 3.3.2. Effects of Different Bacteriostats on DPA Release Rate from Spores

Dormant spores contain a large amount of DPA, and germination of spores releases DPA, which is often used to characterize spore germination [31]. Therefore, in the present study, the effects of different bacteriostats on the germination dynamics of the spores were determined by investigating the release of DPA from the treated spores. As shown in Figure 3, the DPA release steadily increased with increasing incubation time, suggesting the commencement of spore germination. After incubation for 60 min, the released amount of DPA from untreated spores reached 43.52%, while the amounts from spores treated with ethylenediaminetetraacetate, nisin, and tea polyphenols were 20.18%, 22.77%, and 23.22%, respectively, which were significantly lower, indicating that these bacteriostats had good inhibitory effects on spore germination. Among them, *C. perfringens* spores treated with tea polyphenol could germinate, although the number of colonies formed was significantly low, as determined by the plate culture method, suggesting that food additives such as tea polyphenols may play a role in the pathogenesis of *C. perfringens*. The release of DPA after treatment with glutaraldehyde was 19.18% among the four detergents and was consistent with the results of spore turbidity change. In contrast, treatment with hydrogen peroxide Ag+ and 4-chloro-3,5-dimethylphenol did not significantly reduce the DPA release, when compared with that noted in the control group, indicating that these bacteriostats did not considerably inhibit spore germination. The results of the plate counting assay showed that the inhibitory effect of the bacteriostats on the spores was related to the antibacterial activity of these agents and degree of spore germination.

### 3.4. Effects of Different Bacteriostats on Spore Structure during Germination

As the inner membrane of the spores is closely related to spore germination and activity, the effects of bacteriostats on the structure of spores during germination were examined by incubating the treated spores in FTG medium at 37 °C for 1 h and ascertaining the inner membrane damage by measuring the leakage of nucleic acids and proteins at 260 and 280 nm, respectively. As shown in Figure 4, the leakage of spore nucleic acids and proteins treated with tea polyphenol, potassium sorbate, and D-erythorbic acid was significantly higher than that in the untreated group within 1 h of incubation with the germination agent, implying that the membrane permeability might have been altered after bacteriostat treatment, leaking the contents out of the spores during the germination process. Infiltration of tea polyphenol, potassium sorbate, and D- isoascorbic acid into the spores could have promoted changes in membrane fluidity and permeability, causing leakage of more contents from the spores and inactivation of spores. In contrast, treatment with detergents, such as hydrogen peroxide Ag+ and 4-chloro-3,5-dimethylphenol, did not cause a significant increase in the leakage of nucleic acids or proteins from the spores, suggesting that these detergents do not penetrate into the interior of the spores at the initial stages of spore germination, do not affect the spore penetration barrier, and, therefore, do not cause substantial leakage of contents, consistent with the results of DPA release from the spores during germination. Interestingly, glutaraldehyde inhibited germination of spores and prevented leakage of spore contents owing to its sealing effect on the outer layer of the spores. After incubation for 12 h, most of the spores treated with ethylenediaminetetraacetate and glutaraldehyde were phase-bright under phase contrast microscope (Figure 5), indicating that the spores did not germinate completely, further confirming that these bacteriostats prevented the formation of nutrients by inhibiting spore germination [12]. Tea polyphenols, potassium sorbate, and D-isoascorbic acid inhibited germination of some spores by disrupting energy metabolism and acidification, respectively, and also caused further damage to the structure of the budding spores, resulting in the efflux of contents, thus significantly reducing spore survival [34]. Whey protein baking powder neither killed the spores directly nor killed them after germination or prevented the formation of vegetative cells. The other bacteriostats achieved inactivation of spores mainly during the vegetative body formation stage after spore germination and did not cause any significant damage to the spore structure. 

## 4. Discussion

Bacteriostats are essential ingredients in the food industry’s production process because they effectively control foodborne germs and guarantee food safety. Few studies have looked at the impact on *C. perfringens* and its spores, despite several studies demonstrating the effectiveness and mechanisms of bacteriostats against major food-borne infections. Accordingly, in the present study, the inhibitory effects and killing mechanisms of 13 different bacteriostats on *C. perfringens*, a representative strain of wild-type spores, were examined. The spores were studied by comparing the effects of different bacteriostats on spore germination and growth. Nitrite is one of the most widely used food additives in meat production and is an efficient broad-spectrum antibacterial agent. The results of antimicrobial activity (Table 1) indicated that nitrite effectively inhibited the growth of *C. perfringens*, which was consistent with the previous reports [10]. The tea polyphenols, a new natural antibacterial agent, could significantly inhibit the growth of *Escherichia coli*, *Klebsiella pneumoniae*, *Shewanella putrefaciens*, and other microbes [35]; however, they have little inhibitory effect on *C. perfringens*. It might be because *C. perfringens* itself differs from other bacteria in certain ways. The growth of *C. perfringens* was not significantly inhibited by Mauricio’s addition of erythorbate to meat, but it was somewhat inhibited by D-isoascorbic acid [10]. This may be because D-isoascorbic acid caused *C. perfringens* to grow in an acidic environment, which inhibited the growth of bacteria [10,16]. Hydrogen peroxide Ag+ and sodium dichloroisocyanurate + trichloroisocyanuric acid are disinfectants with certain irritations. Studies have shown that the small molecular structures formed after decomposition could penetrate into bacteria and destroy the genetic material in the nucleus, so they are commonly used broad-spectrum disinfectants in industrial production [20]. In addition, the 4-chloro-3,5-dimethylphenol could inhibit the growth and reproduction of *C. perfringens*, and similar results were obtained in *Staphylococcus aureus* and *Escherichia coli.* These results indicated that most broad-spectrum antibacterial agents had certain antibacterial activity against *C. perfringens*. The modes of action of bacteriostats in killing *C. perfringens* spores varied, and the effects were strongly related to the antibacterial activity of these bacteriostats.

In recent years, more and more reports have begun to pay attention to the killing effect of antibacterial agents on spores. It was found that common antibacterial agents could also reduce the number of spores on the plate. Therefore, it was believed that antibacterial agents could also kill spores, as shown in our results (Figure 1). However, it was not clear whether the spores were really dead. Our results demonstrated that fewer spores were able to develop into vegetative forms on the plates after treatment with the antibacterial agents; however, spore turbidity increased and more Ca^2+^-DPA was released with increasing the incubation time, indicating that the antibacterial agents did not directly kill the spores (Figure 2 and Figure 3).

Tea polyphenols have been studied, and an intriguing discovery is that while they do not have great inhibitory effectiveness against *C. perfringens*, they do efficiently prevent the growth of their spores. The rate of the spore germination was not considerably impacted during the early stages, but later on, the spores seemed to stop developing. Spores released more protein throughout the entire germination process. The survival rate of *C. perfringens* spores was greatly decreased by tea polyphenols; it could be primarily attributed to the inactivation of enzymes and disruption of cell metabolism after these substances had penetrated into spores after germination [35]. At the same time, during the spore germination process, tea polyphenols might infiltrate into the spore and act on the spore genetic material, which also increases the possibility that the spore could not grow [36,37]. One theory is that the interior of the spores could become infected with bacteriostats, which could then inhibit respiration and interfere with energy homeostasis, preventing the spores from producing nutrients during later phases of spore germination [27]. While D-isoascorbic acid and potassium sorbate could not stop spore germination, they could decrease spore survival rates and increase protein and nucleic acid leakage during germination (Figure 4). This could be because these bacteriostats also compromised the cell membrane’s integrity and increased cellular permeability, causing the contents of the cell to flow out [9,23,38,39]. The inhibitory effects of nitrite and ԑ-polylysine hydrochloride on *C. perfringens* spores could be attributed to the effect of these compounds on the cell membrane of propagules, enzyme activity, and energy metabolism, thus impairing bacterial metabolism [10,11,40]. Additionally, 4-chloro-3,5-dimethylphenol, a solution of sodium dichloroisocyanurate and trichloroisocyanuric acid, and hydrogen peroxide Ag+ treatments reduced the survival rate of spores. This could be explained by the inactivation of spores and the prevention of nutrient formation after the entry of these substances into the spores after germination [14]. In contrast, whey protein fermentation powder did not exhibit any antibacterial activity against *C. perfringens* spores and could not effectively inhibit spore germination or damage the spores even after their loss of resistance after germination. The antibacterial effect of sodium diacetate is mainly derived from acetic acid. Acetic acid could infiltrate into the spore wall and interfere with various enzymes activities, denaturing proteins. It is worth noting that a number of small compounds exhibited strong antibacterial activities against *C. perfringens*, which may be due to the fact that tiny molecules can more easily penetrate the bacterial cells, alter the formation of compounds associated with bacterial development, hinder the action of enzymes necessary for bacterial life, and destroy bacterial cell structures, instantaneously killing the bacteria [27]. However, spore coatings can prevent the infiltration of macromolecular substances, reduce the infiltration rate of small molecules, and protect the spore cortex [14]. This fully explains why sodium diacetate was able to grow spores of *C. perfringens*. Compounds such as ethylenediaminetetraacetate and glutaraldehyde could prevent spore germination by blocking the spore’s outer layer and inhibiting spore growth by denaturing the protein coat of spores, consistent with other reports [28,41]. In addition, spore germination could alter the permeability of the inner membrane, which may weaken the spore’s ability to withstand chemicals, resulting in the entry of bacteriostats into the spore’s nucleus, leading to inactivation of the spore [14]. Nevertheless, although these findings demonstrated that the inhibition of spores is achieved by penetration of the bacteriostats into the cells after spore germination, the precise stage of penetration of these compounds into the spores, the inhibitory effect, and the specific location of the chemical activity are still unclear.

## 5. Conclusions

Although bacteriostats have been used extensively to eradicate microorganisms, it is unknown how they affect spores. In this study, the effects of various bacteriostats on *C. perfringens* and its spores’ dynamic germination and growth were investigated. The results demonstrated that the bacteriostatic agents primarily prevented spore development in the following three ways: (1) Bacteriostats combine with certain distinct sites in the spore cortex and prevent cortex hydrolysis and inhibit spore germination, keeping the spore in a dormant state for a long time. (2) The germination agents induce hydrolysis of the spore cortex and trigger DPA release. At the same time, the resistance of the spores weakens, and the bacteriostats penetrate into the spores, resulting in decreases in the activity of proteins related to spore germination, changes in the structural components of the spores, and delays in spore germination or inactivation of spores. (3) After spore germination, the resistance of the spores to bacteriostats disappears. The bacterial barrier function is damaged after the entry of the bacteriostats, and the expression and synthesis of intracellular substances such as nucleic acids, proteins, etc., are disturbed, and the normal energy metabolism of cells is affected, leading to impairment of cell respiration and normal growth. Although most of the bacteriostats could not directly kill the spores, they could enter the spores after germination, owing to weakened resistance of the spores, and effectively inhibit nutrient formation and inactivate the spores. These results provide important information on the inhibition mechanism of different bacteriostats on the growth of spores and offer effective insights for the prevention and control of spores to ensure food safety.

## Figures and Tables

**Figure 1 foods-12-01834-f001:**
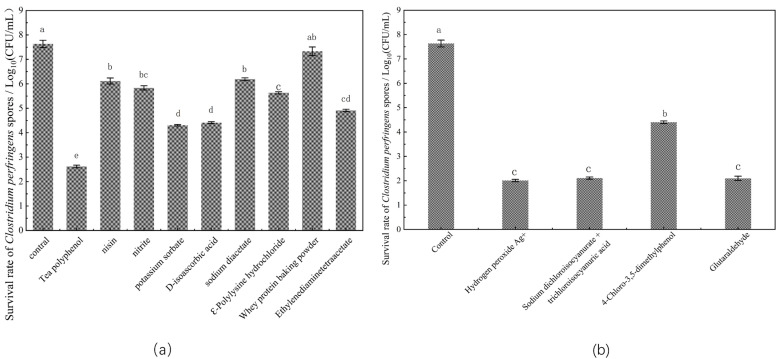
Inactivation of *C. perfringens* spores by different bacteriostats: (**a**) effects of food additives on spore inactivation; (**b**) effects of detergents on spore inactivation. Those with the same superscript letters indicated that the difference was not significant.

**Figure 2 foods-12-01834-f002:**
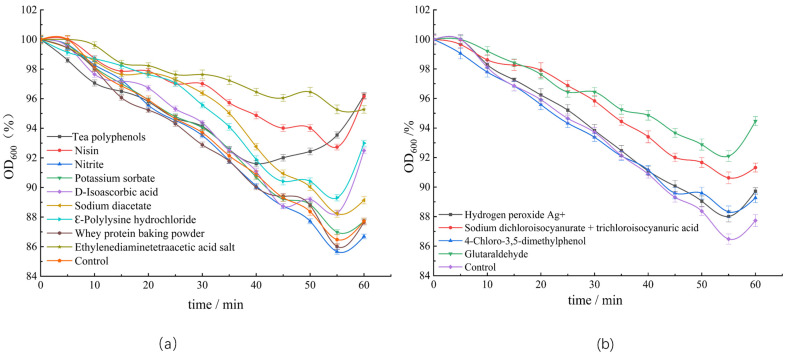
Effects of different bacteriostats on the turbidity of spores: (**a**) effects of food additives on the turbidity of spores; (**b**) effects of detergents on the turbidity of spores.

**Figure 3 foods-12-01834-f003:**
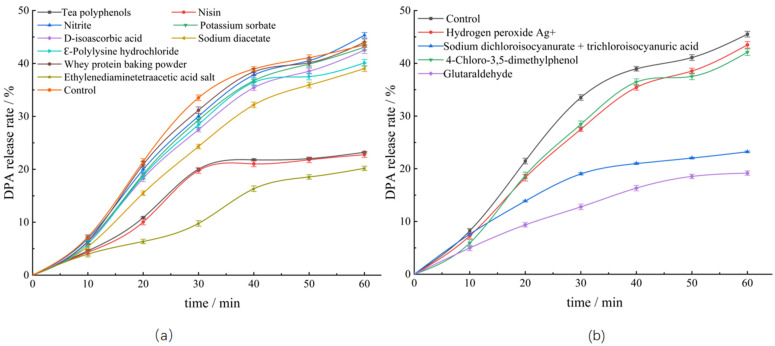
Effects of different bacteriostats on DPA release from spores. (**a**) Effects of food additives on DPA release from spores. (**b**) Effects of detergents on DPA release from spores.

**Figure 4 foods-12-01834-f004:**
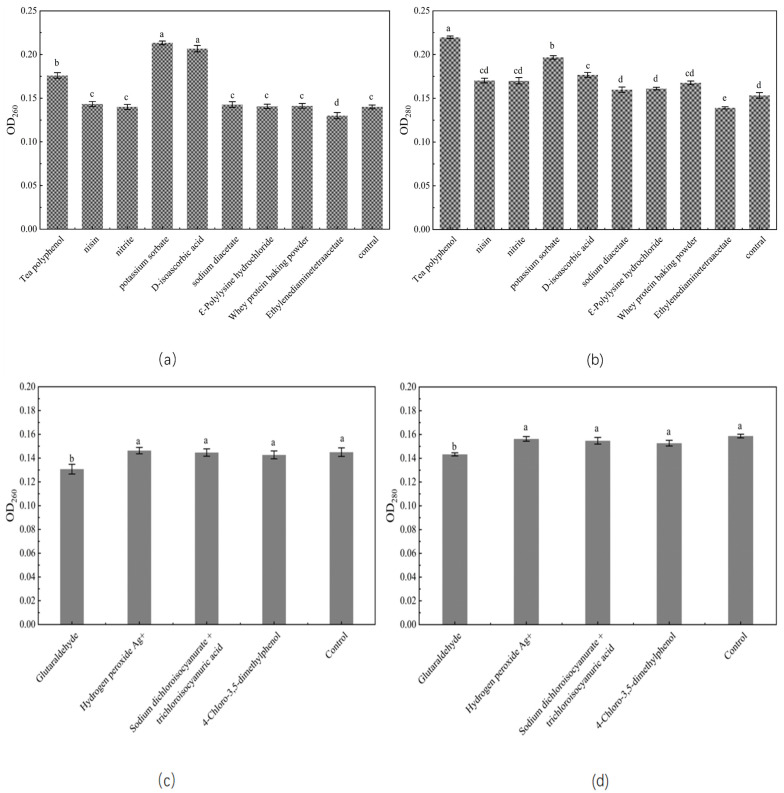
Effects of bacteriostats on the leakage of spore contents. (**a**) Effects of food additives on the leakage of spore nucleic acids. (**b**) Effects of food additives on the leakage of spore proteins. (**c**) Effects of detergents on the leakage of spore nucleic acids. (**d**) Effects of detergents on the leakage of spore proteins. Those with the same superscript letters indicated that the difference was not significant.

**Figure 5 foods-12-01834-f005:**
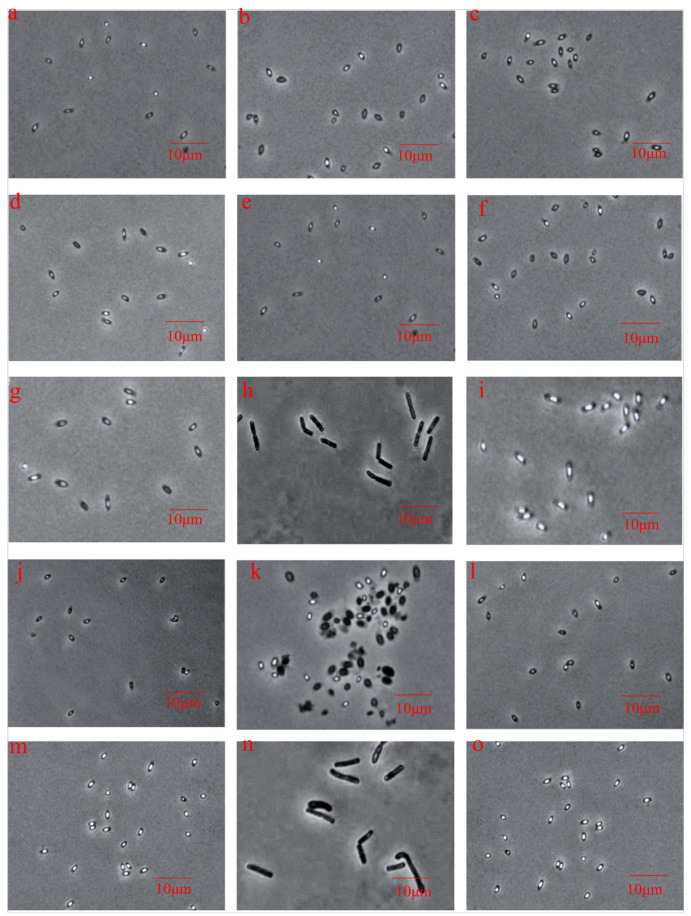
Effects of different bacteriostats on the growth of budding spores. (**a**) Tea polyphenol; (**b**) nisin; (**c**) nitrite; (**d**) potassium sorbate; (**e**) D-isoascorbic acid; (**f**) sodium diacetate; (**g**) ԑ-polylysine hydrochloride; (**h**) whey protein baking powder; (**i**) ethylenediaminetetraacetate; (**j**) hydrogen peroxide Ag+; (**k**) sodium dichloroisocyanurate + trichloroisocyanuric acid; (**l**) 4-chloro-3,5-dimethylphenol; (**m**) glutaraldehyde; (**n**) control; (**o**) non-germinating spores.

**Table 1 foods-12-01834-t001:** Inhibitory effects of different bacteriostats on *C. perfringens*.

Sort	Name of Inhibition Agent	Diameter of Inhibition Circle/mm
Food additives	Control	8.00 ± 0.11 ^a^
	Tea Polyphenols	9.20 ± 0.18 ^ab^
	Nisin	13.67 ± 0.21 ^b^
	Nitrite	32.53 ± 0.98 ^e^
	Potassium sorbate	17.60 ± 0.31 ^bc^
	D-isoascorbic acid	24.67 ± 0.83 ^cd^
	Sodium diacetate	20.20 ± 0.97 ^c^
	ԑ-Polylysine hydrochloride	16.40 ± 0.37 ^b^
	Whey protein baking powder	8.00 ± 0.13 ^a^
	Ethylenediaminetetraacetate	35.33 ± 0.79 ^d^
Cleaning agent	Hydrogen peroxide Ag+	25.57 ± 0.53 ^d^
	Sodium dichloroisocyanurate + trichloroisocyanuric acid	30.00 ± 0.79 ^e^
	4-Chloro-3,5-dimethylphenol	22.57 ± 0.51 ^cd^
	Glutaraldehyde	37.77 ± 0.84 ^f^

Note: Mean ± standard deviation; different superscript letters indicate significant differences (*p* < 0.05).

## Data Availability

Data is not available due to privacy and ethical restrictions.

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
