# Peer review of "Effects of Different Bacteriostats on the Dynamic Germination of *Clostridium perfringens* Spores"

_foods, 2023, doi:10.3390/foods12091834_

Round 1

Reviewer 1 Report

The manuscript: "Effects of Different Bacteriostats on the Dynamic Germination of Clostridium Perfringens Spores" presents important data on the inhibition mechanism of different bacteriostats on the growth of bacterial spores and the results could provide a novel insight for the improvement of food safety and quality.

 The following changes are recommended and some clarifications should be made:

 Material and Methods

 - Please define the abbreviations: TSC plates and FTG medium.

 - How the concentrations of food additives and detergents were selected for this study?

Results and Discussion

 - The section Results should clearly present the data obtained in the study. Thus, some parts of the section Results should be include in the section Discussion. For example this paragraph: "It must be noted that a number of small compounds exhibited strong antibacterial activities against C. perfringens, which may be owing to the fact that tiny molecules can more easily penetrate the bacterial cells, alter the formation of compounds associated with bacterial development, hinder the action of enzymes necessary for bacterial life, and destroy bacterial cell structures, instantaneously killing the bacteria [29]." The same should be done with all paragraph that discussed and compared the present results with other studies. Taking this into account, the sections Results and Discussion should be completely revised.

 - The inhibitor concentrations of the bacteriostats in the Table 1 should be deleted, since they are not variable and were already mentioned in the section Material and methods.

 - There is a typo for the control on the x-axis in Figure 1a.

- Could the authors explain the fact that tea polyphenols exhibited almost the lowest diameter of inhibition zone, but the highest spore inactivation rate?

Author Response

请参阅附件。

Reviewer 2 Report

General comment: The project seems interesting. However, I have found some technical, conceptual, and data analysis gaps in the paper that should be eliminated. I would like to recommend a major revision of the paper. Results and discussion should be combined to help the reader. This is just a suggestion to ameliorate your work.

Please, find my comments, technical corrections and recommendations on the manuscript below:

- Clostridium should be always written in italics.

INTRODUCTION

In the introduction provide more information regarding the bacteriostats used in the work and why they are bacteriostatic agents. Which are the tea polyphenols? Describe also the whey protein baking powder.

It is not necessary to present the experimental design and the achieved results of your study at the end of the Introduction section.

Lines 66 -68: this phrase has to be moved in the conclusion section.

METHODS

- Please provide the full name of all abbreviations (e.g. lines 71, 72, 92, 111, etc..).

- Provide the list of materials used in the work.

- Did you used a tea leaf extract? If yes, did you characterize it in terms of polyphenol content? The activity of an extract is strongly dependent on its composition. Thus, you must provide the extract characterization.

- For each instrument or software, you have to provide information regarding the model, company etc …

- Lines 76, 110, 120: convert the speed of the centrifugation in g value (for reproduction purposes).

- Line 82: polyphenols

- Section 2.2: which solvent did you use to prepare the food additive solutions? Did you use water? If not, why did you use water as blank control at line 95?

Line 93: concentration.

Section 2.4: where did you incubate the spores? Volume?

Line 114: heated at

Section 2.9: which analysis did you use? Post-hoc test?

RESULTS

Lines 139 – 142: add a reference.

Since you prefer to do not combine results and discussion, be careful. As an example, Lines 155 – 159, 174 – 180, 201 – 203, 229 – 234 should be moved to the discussion. There are other sentences that have to be put in the discussion.   

Lines: 164 – 165: move this sentence in the methods.

Line 167: Explain in methods that you transformed your data logarithmic scale.

Fig 1: increase the font size and the quality of the graphs. Insert the statistical results.

Figures 2, 3, 4: increase the font size and the quality of the graphs. Insert in the text the statistical results.

 Line 219: you cannot say that “the number of colonies formed was significantly low” without providing the p-value. The same at line 228. These are just few examples.

Figure 5: insert a scale bar and increase the quality.

DISCUSSION

Your results have to be discussed with other authors’ results. You are not the first authors that used these bacteriostatic.

Author Response

请参阅附件。

Reviewer 3 Report

This study investigated the effects of different bacteriostatic agents on the growth of Clostridium perfringens spores. The researchers examined nine commonly used food additives and four detergents and found that 0.07‰ ethylenediaminetetraacetate had a good inhibitory effect on C. perfringens spore growth, and the spore turbidity decreased by 4.8% after incubation for 60 min. They also found that 0.3‰ tea polyphenols, 0.8‰ D-erythorbic acid, and 0.75‰ potassium sorbate promoted leakage of contents during spore germination. Among the four detergents, 5‰ glutaraldehyde solution presented the best inhibitory effect on the growth of C. perfringens spores, and the spore turbidity decreased by 5.6% after incubation for 60 min.

The study also analyzed the inactivation mechanism of spores by the bacteriostats. The results revealed that bacteriostats could not directly kill the spores, but could inactivate them by inhibiting germination or damaging the spore structure during germination, thus preventing the formation of bacterial vegetative bodies. The researchers compared the leakage of UV-absorbing substances during germination to investigate this mechanism.

Overall, this study provides important information for the mechanism underlying the effects of different bacteriostatic agents on spore growth which could be useful in the development of sterilization protocols for food and other applications. but the authors need to respond to the comments before it goes further

- PLEASE , revise the commas and full-stop of the sentences and check the English of sentences allover the manuscript.

- The first three sentences from the introduction needs to be corrected (not italic just the scientific name )

- the introduction need to be improved While the article does not provide a comprehensive list of specific bacteriostats that were used in the study, it is possible that disinfectants were among the antibacterial agents tested.

- There were a lot of ABBREVIATIONS please revise and in first issue use the complete name then abbreviate it

- please add more details in material and methods section. it is very short to describe what the authors do. it is very confused for the readers

- in line 92 what the mean of that 107–108 CFU/mL. correct and what i know that you could start your trail with one of the concentrations not a range of cfu per ml

-Why did the authors use the Oxford cup in their trail, even though they put only 100 microliters in it? Could the authors include the plate images?

- i wonder how the distilled water could inhibit the bacteria with diameter 8??

- this sign ugly repeated in all the MS "‰"... correct

- All the figures were badly organized need to reconfigure considering abbreviated the treatments in the figure and write the total information in the legend . also increase the resolution and maximize them.

- the discussion part was very bad needs to be more deep and enlarge. please improve and add more references

- remove 6.patents

Author Response

请参阅附件。

Round 2

Reviewer 1 Report

The authors satisfactorily responded to all suggestions.

Author Response

Thank you for your letter and comments concerning our manuscript entitled “Effects of Different Bacteriostats on the Dynamic Germination of Clostridium Perfringens Spores” (ID: foods-2327030). Those comments are all valuable and very helpful for revising and improving our paper, as well as the important guiding significance to our researches.

Reviewer 2 Report

The introduction and the discussion were notably ameliorated. However, the results of the statistical analyses are still missing both in the text and in the figures (see previous comments). Figures were not modified as suggested (the font size must be improved). 

The concentration of the main polyphenols has to be provided since the tea leaf extract's effect is strongly dependent on their content.  

Reviewer 3 Report

I think the manuscript is improved and could be accepted for publication

Author Response

(The authors gave the same response as above.)
